# Sacrificial Layer Technique for Releasing Metallized Multilayer SU-8 Devices

**DOI:** 10.3390/mi9120673

**Published:** 2018-12-19

**Authors:** Anand Tatikonda, Ville P. Jokinen, Hanno Evard, Sami Franssila

**Affiliations:** 1Department of Chemistry and Materials Science, School of Chemical Engineering, Aalto University, 02150 Espoo, Finland; anand.tatikonda@aalto.fi (A.T.); ville.p.jokinen@aalto.fi (V.P.J.); 2Division of Pharmaceutical Chemistry and Technology, Faculty of Pharmacy, University of Helsinki, 00014 Helsinki, Finland; evardhanno@gmail.com

**Keywords:** SU-8, multi layered chips, AZ 15nXT, sacrificial release, microchips, microfabrication, microfluidics, free standing

## Abstract

The low fabrication cost of SU-8-based devices has opened the fields of point-of-care devices (POC), µTAS and Lab-on-Chip technologies, which call for cheap and disposable devices. Often this translates to free-standing, suspended devices and a reusable carrier wafer. This necessitates a sacrificial layer to release the devices from the substrates. Both inorganic (metals and oxides) and organic materials (polymers) have been used as sacrificial materials, but they fall short for fabrication and releasing multilayer SU-8 devices. We propose photoresist AZ 15nXT (MicroChemicals GmbH, Ulm, Germany) to be used as a sacrificial layer. AZ 15nXT is stable during SU-8 processing, making it suitable for fabricating free-standing multilayer devices. We show two methods for cross-linking AZ 15nXT for stable sacrificial layers and three routes for sacrificial release of the multilayer SU-8 devices. We demonstrate the capability of our release processes by fabrication of a three-layer free-standing microfluidic electrospray ionization (ESI) chip and a free-standing multilayer device with electrodes in a microchannel.

## 1. Introduction

SU-8 has gained wide acceptance in micro- and nanofabrication [1] due to its superior mechanical [2] and thermal stability [3] and excellent bio-chemical compatibility [4]. SU-8 excels over most other polymers due to its mechanical properties, e.g., modulus of elasticity of 5 GPa [2] and the ability to easily fabricate high aspect ratio structures (up to 50:1) [5,6,7,8]. It is also easy to fabricate multilayer structures, with multiple spin-coating and exposure steps, and a single development step. SU-8 can be bonded to itself, resulting in channels with four identical walls [9]. There are a wealth of applications of SU-8 in µTAS/Lab-on-chip technologies [10,11]. 

In many cases, free-standing devices are needed. For instance, when sharp tips must be made, as in nozzle and spraying devices, the substrate must be removed. A silicon substrate is also a drawback in applications where high electric fields are used, like capillary electrophoresis. SU-8 is translucent, and sometimes this is used. Two main routes exist for release: sacrificial wafer process, and sacrificial layer processes [12]. Carrier wafer sacrifice is costly and slow, and we do not consider it in this work. 

The basic requirements for a sacrificial layer include easy deposition and patterning, chemical, thermal and mechanical stability during subsequent processing steps, and selective etching in the release step [13]. Many materials have been employed as sacrificial layers: metals such as copper, chromium and aluminum [14,15,16,17], silicon dioxide [9,18,19] and polymers [20,21,22,23]. Polymers include polystyrene, PDMS (Polydimethylsiloxane), SAMs (self-assembled molecules), OmniCoat, PMGI (Polydimethylglutarimide) and various resists such as AZ and un-cross-linked SU-8 itself [15,21,24,25,26,27]. The drawback of inorganic sacrificial layers is the added complexity of thin film deposition and etching. They are often limited in thickness to ca. 1 µm due to slowness of chemical vapor deposition (CVD) and physical vapor deposition (PVD) deposition methods. They are, however, stable and allow different release etch processes to be implemented, e.g., HCl for copper or NaOH for aluminum. Unfortunately they fall short for full wafer release as they generate internal stresses and might lead to micro cracks [24]. Even more important, if the device contains metallization, the release etch might remove the metallization simultaneously. Almost all applications that require metallization of the SU-8 devices use polymer-based sacrificial layers [13,21,28,29].

Polymer-based sacrificial layers can be easily integrated into the microfabrication processes. Spin-coating and development steps are quick and sometimes they can be combined beneficially with the SU-8 processing. Thickness of a spin coated layer can be varied over a larger range than that of CVD or PVD layers. The biggest drawback of polymeric sacrificial layers for metallized multilayer devices is their poor thermal and chemical stability during SU-8 processing [15]. Most polymers are not stable against thermal stresses encountered during metal deposition, resulting in degradation and early releasing of the structure. In [21] it was shown that extreme care has to be taken for metallization integrity on SU-8 structures when using polymer sacrificial layers. OmniCoat, which is a commercially available special polymer developed as a sacrificial layer for SU-8 also falls short, as its not stable when fabricating full wafer multilayer SU-8 microfluidic devices [23,28,30]. Few groups have used common photoresists as sacrificial layers, but they all fall short as these resists are not stable in SU-8 developers for multilayer microfluidic devices, or when very long process times are needed [30]. 

Addressing these issues, we show that the negative photoresist AZ 15nXT (MicroChemicals GmbH, Ulm, Germany) as the release layer allows the fabrication of free-standing multilayer devices with metallization. A wafer scale process is presented, exploiting the inertness and thermal stability of AZ 15nXT in SU-8 processing. The release etch is done TechniStrip NI555 solvent (MicroChemicals GmbH, Ulm, Germany). We show two example processes: fabrication of a three-layer free-standing microfluidic electrospray ionization (ESI) chip and a free-standing multilayer device with electrodes in a microchannel.

## 2. Materials and Methods

### 2.1. Chemical and Materials

Single-side polished 100 mm diameter <100> orientation silicon wafers were purchased from Siegert wafer GmbH (Aachen, Germany). SU-8 negative photoresist (SU-8 50) and developer (mr-DEV-600) were purchased from Microresist Technologies GmbH (Berlin, Germany). AZ 5412E, AZ 15nXT (450 CPS), MIF 312 developer and TechniStrip^®^ NI555 was purchased from Microchemicals GmbH (Ulm, Germany). OmniCoat^™^ and PMGI are purchased from MicroChem Corp. (Westborough, MA, USA). Buffered HF and isopropanol were purchased from Honeywell International Inc., (Kuopio, Finland). Methanol, formic acid, and verapamil for mass spectrometry were purchased from Sigma Aldrich (Espoo, Finland). Polymer sheets used as carriers for bonding purposes were purchased from Arron (Helsinki, Finland). Chrome glass masks for ESI chips were acquired from ML&S GmbH (Greifswald, Germany) and film masks for metalized microchannels were from Micro Lithography Services Limited (Chelmsford, UK). All the reagents for clean room purposes are VLSI grade and for mass spectroscopy they are LC-mass spectrometry (MS) grade.

### 2.2. Microfabrication

Silicon wafers were used as substrates for microfabricating SU-8 multilayers. Resist AZ 15nXT was used as a sacrificial layer underneath the first SU-8 layer. The ESI device contains three layers of SU-8 defined by three lithography steps. The ESI device design and the microfabrication of the first two layers of SU-8 and adhesive bonding of the third layer have been reported previously [31]. All the microfabrication is performed inside a cleanroom and only sacrificial release was done in a chemical hood outside cleanroom.

Multilayer SU-8 fabrication process on silicon is quite challenging due to the large mismatch in thermal expansion coefficients. This generates stresses at the interface, easily resulting in adhesion loss. The problem is compounded when more SU-8 layers are added. To ensure resist adhesion, native oxide was removed from silicon surface in buffered HF solution. Wafers were rinsed in DI water and dried by nitrogen blow gun, followed by baking at 200 °C for a minimum of 15 min to dehydrate the silicon surface. 

SU-8 and AZ 15nXT dispensing was done by manual pipetting. Baking was performed on a programmable hot plate HP-220 from Unitemp GmbH (Germany). Sacrificial layer of AZ 15nXT of 4 µm thickness was spin coated (5000 rpm) on a cleaned silicon wafer and baked at 160 °C for 15 min or soft baked for 110 °C for 3 min and flood UV exposed under mask aligner. The first SU-8 layer (70 µm thick) was applied on top of the cured AZ 15nXT. In our inverted fabrication process this first SU-8 layer will form the roof of the channel in the final device.

For the 70 µm thick (2000 rpm) SU-8 layer the ramping rate of 20 °C/min was used to bring the wafer from room temperature to 65 °C, after which there was a holding step of 15 min. 10 °C/min ramping was then applied to raise the temperature to 95 °C. The hold time at 95 °C was again 15 min. After cooling down to room temperature, the prebaked SU-8 layer was UV exposed (λ = 365 nm) through a photomask with 660 mJ/cm^2^ (22 s exposure) in a Suss MicroTech MA-6 mask aligner. Post exposure baking of the first SU-8 layer is done by ramping from room temperature to 95 °C at 20 °C/min and baking it for 12 min and then cooling down to room temperature at rate of 3.75 °C/min. After cooling down to room temperature, the SU-8 layer was developed in mr-Dev-600 developer for 10 min, washed in isopropanol, and dried. Schematic patterns of different layers are show in Figure 1. 

The second SU-8 layer defines the 50 µm wide microchannel. It is aligned to the sharp tip in the first SU-8 layer. The thickness of the second layer SU-8 layer is 50 µm (3000 rpm) which defines the depth of the microchannel (Figure 1). The prebake ramp rates for the second layer were similar to the first layer, but the hold times were only 5 min at 65 °C and 3 min at 95 °C. Exposure dose was 450 mJ/cm^2^ (15 s exposure). Post exposure bake of the second layer is identical to the first layer. After cooling down the second layer was developed for 10 min, rinsed, and dried as before.

The metallization of the microchannel was performed after the second SU-8 layer processing (Figure 1b). Lift-off patterning used AZ 5412E with standard parameters. Sputtering of 5 nm of Cr and 200 nm Al was done in Plasmalab 400 (Oxford Instruments, Abingdon, UK). The sputtered wafer was immersed in acetone for lift-off (Figure 2b). 

The third layer forms the inlets of the chip. A flexible polymer sheet was cleaned with acetone and isopropanol to remove any residual particles to ensure defect-free SU-8 layer. This polymer sheet was then attached to a silicon carrier wafer with double-sided tape. The third layer of SU-8, 70 µm thick, was spin coated. Prebaking conditions were identical to first layer of SU-8 except the cooling step, which was at a rate of 3.2 °C/min from 95 °C to 63 °C and then held at this temperature for 20 min. Simultaneously the silicon wafer with 15nXT and the first and the second SU-8 layer was heated to 63 °C. The polymer sheet was then released from the tape on the carrier wafer and bonding SU-8 layer is placed carefully on top of the second layer of SU-8 and laminated with minimal force to improve adhesion between the layers. This leaves the non-exposed 3rd SU-8 layer under polymer sheet in contact with the exposed 2nd layer. The wafer stack was kept on hot plate at 63 °C for 15 min and then freely cooled to room temperature. The third layer was exposed with dose of 750 mJ/cm^2^ (25 s exposure) through the polymer sheet. After post bake the polymer sheet was peeled off. Development of the 3rd layer was identical to the 1st layer. The ESI chips (Figure 2a) are of 2.2 cm × 3 cm and the metalized chips (Figure 2b) are 2 cm × 2.5 cm with both having channels of 50 µm deep and width.

### 2.3. Cross-Linking and Release Etching

Sacrificial layer AZ 15nXT was used for releasing multilayer SU-8 microfluidic devices from the silicon wafer. For all the chip designs we used 4 µm thick AZ 15nXT. We have used two different methods for cross-linking of the AZ 15nXT. AZ 15nXT, being a negative photoresist, is cross-linked by UV exposure or by heating at elevated temperatures. The UV exposure was performed by flood exposure of 2800 mJ/cm^2^ in the same MA-6 mask aligner as the SU-8 exposures, and the thermal curing was done at 160 °C on a hot plate for 15 min. 

For releasing, we demonstrate three different sacrificial etch techniques. In first method the SU-8 processed wafer was immersed in room temperature TechniStrip NI555 for overnight. In the second method, TechniStrip was heated to 60 °C. The third method involved a thermal shock process where the processed wafer was rapidly heated to 120 °C on a hot plate for 20 s and placed immediately in TechniStrip solution which was held at RT. The holding time of the wafer in Technistrip was determined by waiting until the chips start to float to the top of solution. We also compared the sacrificial release of SU-8 layers with commercially available polymer materials OmniCoat and PMGI. For these comparison tests we have spin coated, baked and exposed OmniCoat and PMGI according to parameters provided by MicroChem Corp. [32,33].

### 2.4. Mass Spectrometry Measurements

Agilent 6300 Series Ion Trap (Santa Clara, CA, USA) was used for these experiments. Capillary voltage of −1600 V was applied and mass range of 100 to 600 m/z was scanned. Drying gas temperature was set to 25 °C and flow set to 3 L/min. The SU-8 electrospray chip was placed onto the XYZ stage and positioned so that the distance between electrospray tip and the cone was 2–3 mm. A platinum electrode connected to a high voltage source (built in-house) was placed into the solvent well of the chip. 1500 V (compared to ground) was applied to the electrode after which 2 µL of spray solvent (90% MeOH, 10% Milli-Q water (Merck Millipore, Burlington, MA, USA), which both contained 0.5% formic acid) was added to the well. In the case of verapamil measurements, 400 nM solution of the analyte in spray solvent was prepared from verapamil hydrochloride. 

## 3. Results and Discussion

To have a free-standing embedded microfluidic channel made entirely of SU-8, we need three layers of SU-8, patterned on top of each other, aligned and bonded together [19]. The sacrificial layer is the most critical and challenging step, as it determines the release etch. The addition of metallization adds its own peculiarities as it limits the kinds of sacrificial layers and etchants that can be used.

The rationale behind using the AZ 15nXT as sacrificial layer was that it is a widely used resist for electroplating. For this purpose, AZ 15nXT is inert for acidic condition and cyclic electrical fields which are encountered during electroplating process. Therefore, AZ 15nXT crosslinks into highly stable state, which also makes it resistant against most common developers and solvents used in microfabrication. Stripping AZ 15nXT is accomplished by a specifically selected solvent TechniStrip^®^ NI555 which is a mixture of acidic-based solvents [34]. The stability of AZ 15nXT depends on the extent of cross-linking. Once the AZ 15nXT has been cross-linked by either the UV radiation or thermal means, it is highly stable over wide range of conditions. Specifically, it does not dissolve or swell in solvents of SU-8 process or mr-DEV-600 a SU-8 developer. This is different from other common resists which have been used as sacrificial layers before [15,25]. For ESI devices the total development time for three SU-8 layers was 30 min and for metalized chip we have total 33 min development times. The recommended soft bake for 15nXT is 3 min at 130 °C. However, during multilayer SU-8 processing, thermal cross-linking was done at 160 °C for at least 15 min. Due to high degree of cross-linking by UV, nXT is stable during the metal deposition process. UV-assisted cross-linking of AZ 15nXT slows the dissolution in TechniStrip, for this reason UV cross-linking of AZ 15nXT was used only when metallization of SU-8 devices was needed (Figure 1b). Thermal cross-linking of AZ 15nXT was done by baking the resist above its post bake temperature. We have tried temperatures and times ranging from 120 °C to 200 °C for times 3 min to 30 min and found 160 °C and 15 min baking to be optimal for our process. This is based on stability of the AZ 15nXT in multiple SU-8 developing steps and the ease of releasing in Technistrip NI555. AZ 15nXT exhibited excellent adhesion to silicon and SU-8 layers as well as high stability in SU-8 developer after cross-linking (Figure 3a and Figure 4a,b).

In the first release method (Method 1), the whole silicon wafer with the SU-8 multilayer structure (with or without metallization) was immersed in TechniStrip NI555 overnight. AZ 15nXT slowly swells and dissolves in TechniStrip. TechniStrip also penetrates the underlying nXT releasing individual chips. However, the rate of dissolution of the AZ 15nXT in TechniStrip is slow making the release process time consuming. 

In the second method (Method 2), we used heated TechniStrip at 60 °C. This releases SU-8 chips in less than 3 h. The silicon carrier wafer released by both methods one and two show very little residues of AZ 15nXT, making them reusable with minimal cleaning (Figure 3b,c).

To release the chips much faster we have developed a third method (Method 3), which is based on thermal shock release of AZ 15nXT. In this technique, the silicon wafer with SU-8 layers was heated on a hot plate for 20 s at 120 °C and placed immediately into TechniStrip solvent which was held at room temperature. Within 15 min individual SU-8 chips were released from the silicon substrate (Figure 3b,d). Due to thermal expansion difference between silicon (2.6 ppm/°C) and the SU-8 (52 ppm/°C), thermal stresses developed at the 15nXT layer between silicon and the first SU-8 layer. Thermally assisted release process leaves more contamination on the silicon wafer, which can be removed by incubating the released silicon wafers in TechniStrip followed by Piranha cleaning. 

According to the MicroChem corp. it takes 20 min for AZ 15nXT to swell and completely dissolve in the TechniStrip solution. These are for cross-linked AZ 15nXT from the manufacturer conditions and it takes much longer for relatively heavily cross-linked AZ 15nXT and would need heating of the TechniStrip or even sonication of the solution during release, which might damage microfabricated devices [34]. However, the processing conditions required for multiple layers of SU-8 modify the resist, making it much harder to remove. The exact swelling/dissolving rate is not known, but it was at least stable for more than 30 min in SU-8 developers. 

OmniCoat™ is recommended by the MicroChem corp. as a sacrificial layer for SU-8 microfabrication [33]. It is ideal as a sacrificial layer for microfabricating single layer of SU-8 or multilayers without any sealed or embedded structures but OmniCoat falls short when used for multilayer processes [30]. It was reported earlier that OmniCoat has a poor adhesion with the silicon and SU-8 making resulting in higher stresses and delaminating the SU-8 structures [35]. This is ideal for single layer SU-8 whereas for multilayers it falls short, as the stresses would delaminate during the processing. PMGI is a lift of resist which is commonly used for sacrificial release in MEMS [36,37,38]. Thermally cured PMGI is not suitable for multilayer sacrificial release and needs deep-UV exposure to make it inert for SU-8 developer [38]. 

### 3.1. Comparison between AZ 15nXT, OmniCoat and PMGI

We have compared our AZ 15nXT release process with OmniCoat™ and PMGI, which are commercially available polymer sacrificial layers for SU-8 process. The first layer of SU-8 was spin coated, soft baked, exposed, and post baked identically on all three-release layer materials. OmniCoat and PMGI wafers were placed in SU-8 developer and within minutes, individual chips were released from the silicon wafer (Figure 4c,d). This suggests that OmniCoat and PMGI are not stable in SU-8 developer and are only suitable for single layer of SU-8 process. 

We tried to optimize release layer thickness, exposure and baking temperatures but could not improve OmniCoat or PMGI for multilayer process. The sacrificial layer encounters multiple solvent and thermal steps during microfabrication. Since single release etch step remains the most viable option for practically all devices, the sacrificial material must tolerate multilayer processes. AZ 15nXT is robust and superior to any other polymer-based sacrificial release layer for multilayer SU-8 structures. The stability of 15nXT can be tailored easily to suit any process by simply changing the thickness, baking time or UV exposure making it adoptable to other multilayer applications. Sacrificial etching of long channels from the mouth of the channel is not a viable option because of excessive etch times (overnight or several days) and SU-8 deformation [18]. 

Schmidt et al. [28] have shown that OmniCoat can be used for SU-8 multilayer process in which sacrificial release was performed using common clean room developer MIF-319. This limits the selection of photoresists that can be used for microfabrication and making the metallization process much more challenging. Valencia et al. [37], and Bohl [30] have shown that PMGI and OmniCoat respectively can be used for processing SU-8 multilayers where all the SU-8 layers are developed together in one single step. This is not suitable for embedded microchannel as it would take long development time for SU-8 in embedded micro channels [18]. Foulds et al. [38] have shown that PMGI can be used as sacrificial layer for making free-standing multilayer SU-8 MEMS. The biggest problem arising was the cracking of PGMI itself resulting in defects in the SU-8 layers and other lithography process which are microfabricated on PGMI surface. 

Polymer-based sacrificial layers which have been developed for MEMS applications where the areas to be released are fairly small, usually device element size, but in microfluidics release is often about full chips or even wafers. Because of this difference, sacrificial materials which are good for MEMS, such as PGMI [38], cannot be used for microfluidic devices (Figure 4d). This effect is more pronounced when we have multilayer SU-8 process. 

### 3.2. ESI Chip

To test the stability of thermal cross-linked AZ 15nXT sacrificial layer we have microfabricated multilayer SU-8 microfluidic device for MS. Free-standing SU-8 chips are used for detecting biomarkers by MS. The chip contains application reservoir, a microfluidic channel, and a sharp tip for spraying. A sharp tip is needed to establish a stable Taylor-cone. The droplets are ionized either by a corona needle or an UV-lamp. The functionality and performance of these free-standing SU-8 ionization chips were evaluated by analyzing molecules of verapamil by mass spectrometry. The limit of detection (LOD) was measured with injection under Selective Reaction Monitoring (SRM) using two pre-selected precursor/product ion pairs of *m*/*z* 455→165 and 455→303. The LOD we have achieved was 3.62 nM with a good correlation factor of 0.97 (Figure 5b). These are similar or even better to what we have reported with similar SU-8 multilayer devices [25,26,36,38]. We speculate that this is due to reduced etching times and complete dissolution of the sacrificial layer, i.e., fewer residues. SU-8 devices released with the AZ 15nXT as sacrificial layer behave consistently compared to the devices which are released by other methods [18,19,31,39]. 

### 3.3. Free-Standing Metallized Chip

To test the stability of AZ 15nXT as sacrificial layer in metallization process, we have used UV cross-linked AZ 15nXT. After the patterning of the first and second layer of SU-8, we have metallized the microchannel with Cr/Al (5 nm/200 nm) (Figure 1b). After lift-off patterning of the metal electrodes on the second SU-8 layer, we bonded the lid of the microchannel as before, and released the chips by the first method (Method 1) in TechniStrip (Figure 2b). For the metallization process, thermally cross-linked AZ 15nxt was not stable in sputtering process. The UV cross-linked AZ 15nXT could withstand the thermal stresses caused by the sputtered layers and handle the process solvents in the metal patterning step. Inorganic sacrificial layers such as metals and oxides are not compatible for microfabrication of metallized structures. Acidic process such as HF [9,19] are needed for oxide releasing or HCl for metal release, and these adversely affect the integrity of the metalized SU-8 structures. This limits the choice of sacrificial layers to organics for metalized SU-8 chips. This section may be divided by subheadings. It should provide a concise and precise description of the experimental results, their interpretation as well as the experimental con clusions that can be drawn.

Based on the optimization, we propose the following two optimized processes to be use for residual-free release and fast release of SU-8. For residual-free release the best method was to spincoat 4 µm of AZ 15xNT and bake at 160 °C for 15 min followed by process the multilayer SU-8 devices on AZ 15nXT and place the processed silicon wafer in TechniStrip solution at 60 °C during the release (Method 2). Thermal shock (Method 3) is best for fast release of the SU-8 chips process as before. The extent of AZ 15nXT stability depends on the thickness of the film, UV exposure dose and time, and the bake temperature. We have optimized all these parameters to achieve highly stable AZ 15nXT sacrificial layer for our multilayer SU-8 process at the same time keeping in mind the ease of full wafer release of the devices in TechniStrip.

## 4. Conclusions

We have developed a novel fabrication process where negative resist AZ 15nXT serves as a sacrificial layer for multilayer SU-8 process. Using AZ 15nXT resist will significantly reduce the processing steps needed to make released devices. For residual-free release of multilayer SU-8 devices the best method is to place the processed silicon wafer with SU-8 structures in heated TechniStrip solution and for fast release thermal shock method is the best solution. We have demonstrated the performance of the released SU-8 devices in mass spectrometric analysis of verapamil with reproducible spraying and low limit of detection.

AZ 15nXT is a good compromise between stability during device processing and reasonably fast release. While OmniCoat and PGMI can be used to release simple SU-8 chips, our process is suitable for any geometry and any number of SU-8 layers. Our general release method will help in stable integration of metallization steps into the polymer MEMS and can readily be used for other microfabrication process.

## Figures and Tables

**Figure 1 micromachines-09-00673-f001:**
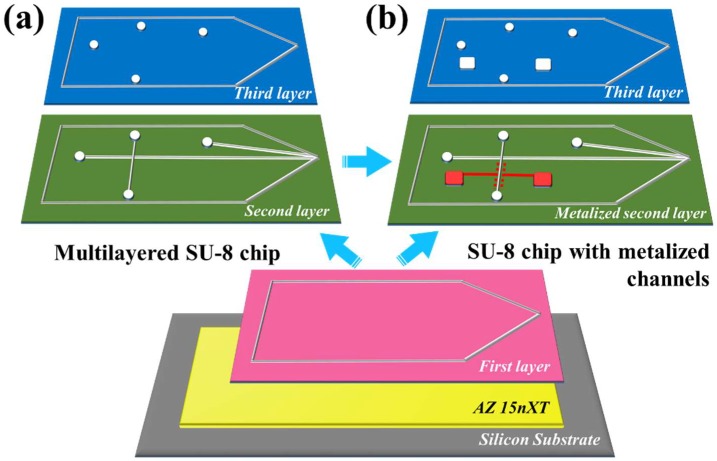
Schematics of multilayer SU-8 free-standing microfluidic device for mass spectroscopy electro spray ionization. (**a**) the multilayered SU-8 ESI chip (**b**) SU-8 chip with extra metallization step.

**Figure 2 micromachines-09-00673-f002:**
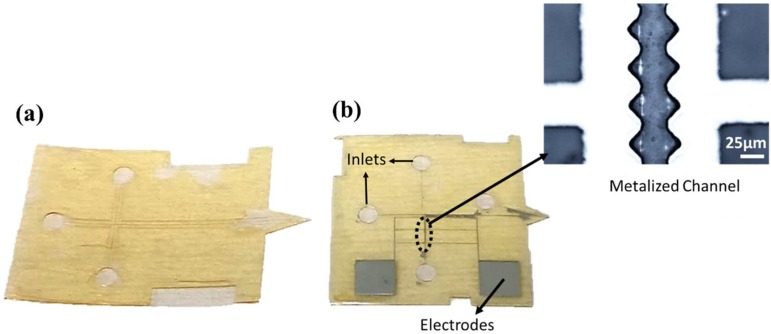
Microfabricated multilayer SU-8 chips (**a**) the multilayered SU-8 electrospray ionization (ESI) chip for mass spectrometer analysis (**b**) metalized SU-8 chip with electrodes. The insert shows the channels which are metalized by lift-off.

**Figure 3 micromachines-09-00673-f003:**
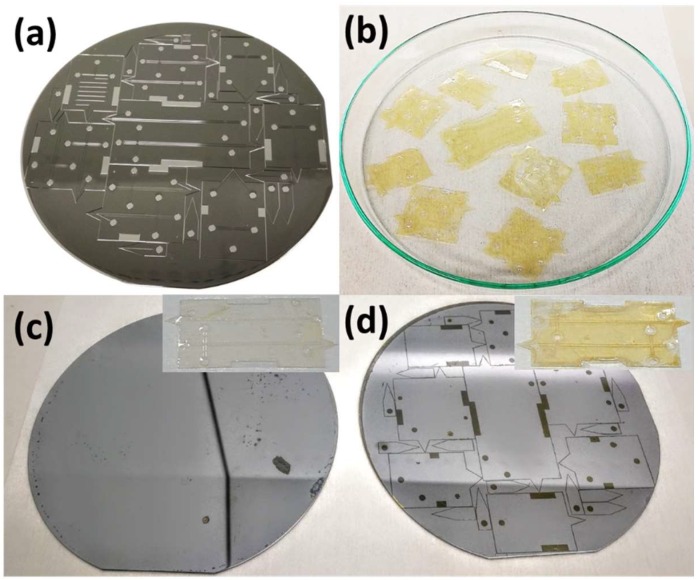
(**a**) Multilayer SU-8 devices fabricated on AZ 15nXT on silicon wafer before release (**b**) Free-standing SU-8 ESI chips after releasing from the substrate (**c**) Silicon wafer after releasing by wet Technistrip assisted release (Method 1 or 2), which leaves very little residue and can be reused with simple cleaning steps (**d**) silicon wafer after releasing from thermal assisted Technistrip release (Method 3), which leaves more residues. Insert shows the released chips. Residual AZ 15nXT from chips can be removed by incubating for more time in Technistrip.

**Figure 4 micromachines-09-00673-f004:**
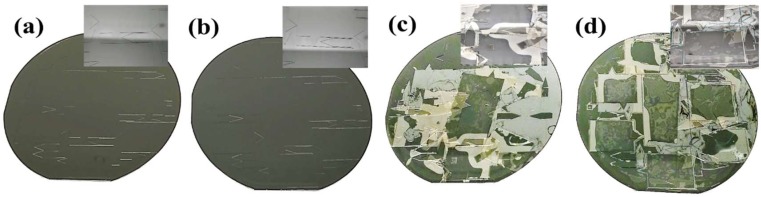
Multilayer SU-8 devices fabrication, during the first layer development (**a**) AZ 15nXT thermally cross-linked (**b**) AZ 15nXT UV cross-linked (**c**) Omni coat and (**d**) PMGI in place of AZ 15nXT. The SU-8 surfaces have delaminated from the silicon substrate when we have OmniCoat or PMGI are used as sacrificial layers making the multilayer process impossible. The first layer is quite stable when we have used AZ 15nXT as sacrificial layer. Insert shows the zoomed in section of the wafer where in (**a**,**b**) the pattern is intact in (**c**,**d**) the patterns have peeled off from the surface.

**Figure 5 micromachines-09-00673-f005:**
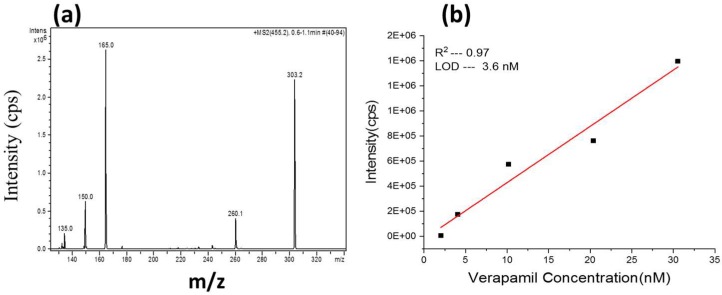
MS spectra for Verapamil (**a**) internal standards were protonated verapamil at m/z 455.3 (**b**) Limit of detection for S/N of 2.5 was 3.6 nM with correlation coefficient of 0.97.

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
