# Peer review of "Sacrificial Layer Technique for Releasing Metallized Multilayer SU-8 Devices"

_micromachines, 2018, doi:10.3390/mi9120673_

Round 1

Reviewer 1 Report

This manuscript presents the novel microfabrication technique where the photoresist (AZ 15nXT) can be utilized as a sacrificial layer during the practical multilayered SU-8 device fabrications. The manuscript is overall well-written and easy to read, with sound presentation of methods and findings. The manuscript could be further improved to the publishable level by addressing several minor issues suggested as followings:

1. Additionally specifying UV exposure time would be helpful to understand the experiment, for instance, in between lines of 102-110.

2. The resolution of Fig. 2(b) could be enhanced for better readability.

3. Adding the enlarged pictures showing the differences of (a) vs. (b), and (c) vs. (d) in Fig. 4 could be great for readers' better understanding.

Author Response

1.       Additionally specifying UV exposure time would be helpful to understand the experiment, for instance, in between lines of 102-110.

We have added the exposure time in the manuscript as suggested.

2.       The resolution of Fig. 2(b) could be enhanced for better readability.

Image was modified as suggested

3.       Adding the enlarged pictures showing the differences of (a) vs. (b), and (c) vs. (d) in Fig. 4 could be great for readers' better understanding.

Image was modified as suggested

Reviewer 2 Report

Journal: Micromachines(ISSN 2072-666X)

Manuscript ID: micromachines-399141

Type of manuscript: Communication 

Title: Sacrificial layer technique for releasing metallized multilayer SU-8 devices  

The manuscript describes unique approach for the fabrication of stand-alone type microfluidic device made of SU8 which is well-known and commercially available photosensitive polymer material in the field of Lab-on-a-tip. The authors discussed new approach to remove the fabricated microfluidic device from the silicon wafer substrate using sacrificed layer. This manuscript contains unique and potentially useful for some applications in the general SU-8 device fabrication technique. The paper provides very interesting data but it still needs major revision to be acceptable for the Journal “Micromachines” as follows.

1.      The Authors described the holding time of sacrificed layer in a certain device fabrication process. However, physical etching rate of sacrificed layer (4µm of AZ 15nXT) at the reference environment is not discussed. The etching rate of AZ 15nXT depends on the design of device structure and thickness of sacrificed layer. This should be carefully described in the manuscript as an academic research.

2.      It is not clear the detail of the experimental setup of development process. It is necessary to describe the detail how to determine the holding time?

3.      What does thermally cross-linked AZ 15nXT mean? What is the difference from soft bake written on the specification sheet of AZ 15nXT?

4.      In the mass spectrometry measurements, is there any residual AZ 15nXT on the surface which contacted to AZ 15nXT on one of the first SU8 layer?

I hope these comments will be helpful for the improvement of the manuscript.

Author Response

1.          

1.           The Authors described the holding time of sacrificed layer in a certain device fabrication process. However, physical etching rate of sacrificed layer (4µm of AZ 15nXT) at the reference environment is not discussed. The etching rate of AZ 15nXT depends on the design of device structure and thickness of sacrificed layer. This should be carefully described in the manuscript as an academic research.

It takes 20 minutes to swell and dissolve a 4µm layer according to the manufacturer’s datasheet. However, the processing conditions required for multiple layers of SU-8 modify the resist, making it much harder to remove. The exact swelling/dissolving rate is not known, but it was at least 30 mins stable in the SU-8 developers when cross linked thermal means and even longer when cross linked by UV.

We have added more information regarding this in the  revised manuscript (lines 254-260)

In this work we are releasing relatively large microfluidic chips, whose dimensions are 2.2 cm x 3 cm(page 4). We also added this information to the results and discussion to make it explicit that it is to this chip size that our release time results refer to.

2.           It is not clear the detail of the experimental setup of development process. It is necessary to describe the detail how to determine the holding time?

The SU-8 wafers are developed by directly placing the wafers in the mr-Dev-600 developer solution and incubating them for a fixed time which depends on the SU-8 layer thickness. For metallized chips we have an extra development step in MF312 developer for 3 mins. For ESI devices the total development time for three layers was 30 mins and for metallized chip we have total 33 mins. AZ 15nXT has shown superior stability in all these development steps. We have emphasized this in the revised manuscript (Added extra sentences in line 203-206 to be very clear)

In case the reviewer was referring to the 15nXt release process instead of the development steps in the SU-8 processing, we have clarified this in the experimental on page 4 and 5 in section 2.3 Crosslinking and release etching. The holding time of the wafer in Technistrip was determined by waiting until the chips start to float to the top of solution. (Added extra sentences in line 172-173 to be very clear)

3.           What does thermally cross-linked AZ 15nXT mean? What is the difference from soft bake written on the specification sheet of AZ 15nXT?

AZ 15nXT is a negative resist which can be cross linked either by UV radiation or thermal means at elevated temperatures. For thermal crosslinking the baking temperature should be higher than the hard bake temperature, or it should be very long hard bake time than the specified from the company (MicroChemicals GmbH). The hard baking temperature of 15 nXT is  130oC and 3 mins. We have tried temperatures and times ranging from 120oC  to 200oC for times 3 min to 30 min, and found 160oC and 15 min baking to be optimal for our process.

This clarification has been added to lines 203-206

4.           In the mass spectrometry measurements, is there any residual AZ 15nXT on the surface which contacted to AZ 15nXT on one of the first SU8 layer?

We have not observed any contamination of AZ 15nXT specifically. As fabricated our device in inverted form i.e the inlets are in the 3rd layer resulting in very low contamination from the sacrificial layer.  In order to make sure we don’t have any residual contamination the released chips are incubated for 3 more hours in the Technistrip.

Reviewer 3 Report

The results in this manuscript are useful to the micro-fabrication community. It can be published in Micromachines after an optional revision:

1.       Fig. 3(b) shows samples fabricated by both methods 2 and 3. An enlarged view of samples is helpful. Or a label of sample linked to method 2 or method 3 will help understand the quality difference.

Author Response

        Fig. 3(b) shows samples fabricated by both methods 2 and 3. An enlarged view of samples is helpful. Or a label of sample linked to method 2 or method 3 will help understand the quality difference.

The samples from 3(b) are released by method 3 and samples which are released from method 2 were not shown here as they look similar. The enlarged chips are shown in Figure 2. But as suggested we have added inserts to image 3.

Round 2

Reviewer 2 Report

Dear Authors,

the manuscript is well revised.

I recommend to accept after following minor revision.

There is a typo in line 210.

15nxt --> 15nXT

Yours sincerely,